# Superimposed Renewal of Industrial Heritage under the Guidance of Low Maintenance and Sustainability—Renewal of Refinery Site in Jinan Tianhong Community

**Zijia Li [1,2], Qiyu Gai [2,3] and Luofeng Qin [1,*]**

[1] College of Civil Engineering and Architecture, Zhejiang University, Hangzhou 310058, China; licicalizijia@zju.edu.cn

[2] Center for Balance Architecture, Zhejiang University, Hangzhou 310058, China; 22016260@zju.edu.cn

[3] College of Agriculture and Biotechnology, Zhejiang University, Hangzhou 310058, China

[*] Correspondence: qinluofeng@163.com

**Abstract:** The renewal of industrial heritage is a long process. With the development of society, a single form of renewal can no longer meet the needs of the public and environment in China. In the case that a large number of industrial heritage sites require secondary renewal, it becomes an urgent issue to consider how to achieve sustainability in the process of superimposed renewal, reduce the amount of future renewal changes, and at the same time realize economic and environmental friendliness, reduce maintenance costs, achieve cyclic spontaneous renewal, and ensure the feasibility, variability and growth of the reserved renewal. Jinan Tianhong Community Refinery Industrial Heritage Park was selected as the case study in this paper based on the theory of low maintenance to explore the strategy of superimposed renewal under the guidance of low maintenance and sustainability-oriented superimposed design in the social process; thus, the design of the secondary renewal of industrial heritage can create an eco-friendly space for activities from an ecological perspective, so as to achieve low-maintenance, low-intervention and sustainable industrial heritage renewal in the long-term.

**Keywords:** low maintenance; sustainability; industrial heritage; superimposed design

## 1. Introduction

With the passing of the era of rapid construction and renewal, industrial heritage has gradually revealed some problems and drawbacks that were overlooked during its first renewal after more than 10 years of use. Some of these sights neglect the environmental elements and the issue of historic succession, and their construction methods and spatial characteristics no longer meet the needs of people's lives and are not in line with the general trend of sustainability. If we continue using the renewal method of continuous renewal and reconstruction, we will not meet the requirements of environmental friendliness.

Low-maintenance landscapes are landscapes with artistic, social, ecological and cultural values at the lowest possible cost (resource cost and environmental cost) [1]. Many industrial heritage sites in China lacked low maintenance and low intervention in the process of their first regeneration, and there is a shortage of post-use evaluation and reflection on the built environment after such regeneration. Therefore, there is an urgent need to introduce the strategy of 'low maintenance sustainable superimposed renewal' at the critical point of the new round of renewal. 'Low maintenance sustainable superimposed renewal' refers to the second or successive renewal processes that take sustainability as the principle, consider low maintenance as the design method and guidance, and take the long-term development vision of the base into account. This promotes the possibility of future regeneration without overturning the whole thing, and a sustainable renewal process.

The Tianhong Community Children's Park was selected for the case study, which was converted from the Jinan Oil Refinery site, a typical first renewal project that gradually assumed a negative trend in the course of urban development. This study investigated and conceptually designed the superimposed regeneration of the site, using it as an example to summarize the strategies of superimposed regeneration under the direction of low maintenance sustainability. The study aimed to explore how to reserve space for renewal in the current situation where the era of rapid construction and renewal has passed and the results of initial renewal are gradually failing so that in the case of industrial heritage facing secondary renewal and multiple renewals, sustainable, spontaneous and low intervention development of superimposed renewal can be achieved. The goal is to make the process of spontaneous renewal of industrial heritage compatible with the speed of social and urban development and to reduce the waste of resources and labor costs associated with rebuilding.

The object of the study is the former oil refinery in Tianhong Community, Licheng District, Jinan, which covers an area of about 50,000 m². After the first renewal, it was transformed into a children's park. The existing industrial heritage children's park, which was renovated for the first time, is no longer able to meet the growing demand of the residents and the environmental needs guided by the positioning of the city as a 'spring city' because of the rapidly rising urban areas and traffic routes around the base, which have led to an increasing number of residential areas. Therefore, an environmentally friendly form of superimposed secondary renewal is imperative, to transform it into a community park serving the residential areas that are constantly being built around the base. The main issue facing the regeneration of this industrial heritage is also the main issue addressed in this study. The site is crossed by an urban sub-road in the course of urban development, and the main issue investigated in this study was the development of a sustainable design approach to maintain the integrity of the separated site after secondary regeneration, and the harmonious integration of the two large, abandoned oil tanks in the south of the untreated site with the park in the north (Figure 1). The historical and human values of the industrial heritage were recognized and explored and it can be utilized in the future, and achieving a low-maintenance sustainable overlay regeneration was also a point of inquiry in this study.

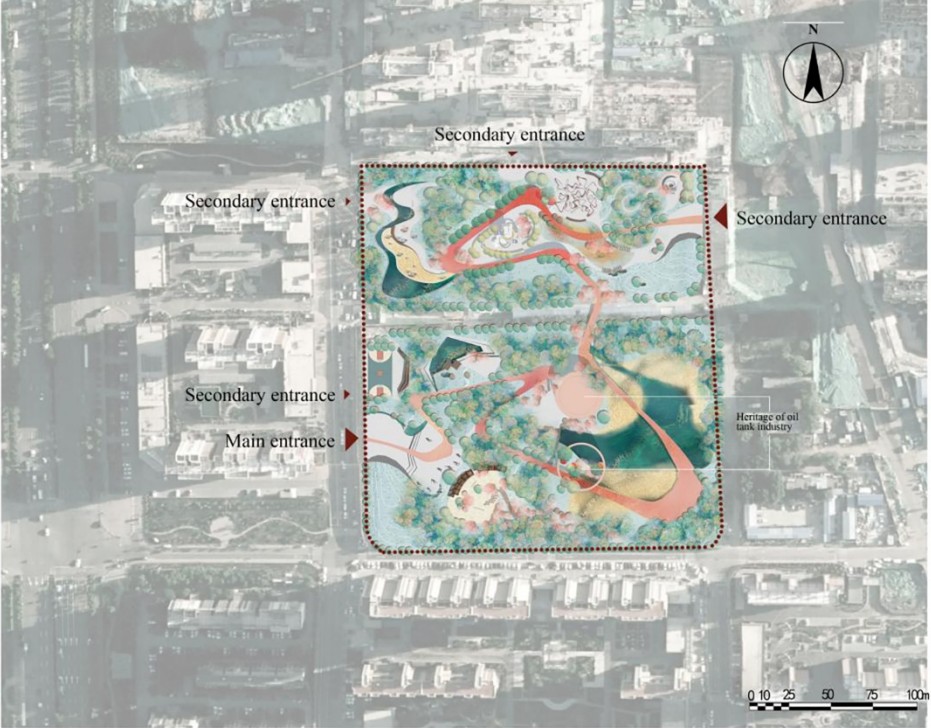

**Figure 1.** The second renewal design of site (drawn by authors).

## 2. Literature Review

### 2.1. Research on the Reuse of Industrial Heritage

Industrial heritage is the remains of industrial culture with historical, technical, social, architectural or scientific value [1]. Industrial heritage not only has substantial architectural heritage attributes, but also historical attributes that record changes in labour patterns over time. Industrial heritage defines locations where human beings were engaged in industrial production, reflecting the production scenes and technologies of the time and carrying the historical memory and cultural deposits of the industry and the city.

In the middle of the twentieth century, the world gradually began the trend of transformation from the industrial age to the information age, and cities entered into chaotic and disorderly expansion. A large number of industrial remains were distributed in the original urban centers, urban waterfront areas and suburbs, including a large number of factory buildings and industrial areas. Since the buildings themselves have not reached the end of their architectural life, the treatment and reuse of a large amount of industrial remains from the past decades becomes a key issue. Their adaptation and reuse can effectively reduce the generation of urban waste and excessive waste of resources, which has important economic and ecological significance for achieving sustainable development.

Following the selection of the first World Heritage Site in 1978, 1154 properties were included on the World Heritage List (Resource: https://whc.unesco.org/en/list/?search= &order=country (accessed on 10 May 2022)) by the end of 2021. Based on the World Heritage List and the World Heritage Centre's rubric for heritage items, and also with reference to the research findings of Du Biebei, Duan Yong [2], Cui Weihua and Gong Lina [3], Cui Weihua, Wang Zhiyu and Xu Bo [4], the recognition of heritage sites by year is plotted in Figure 2. Due to the year-on-year increase in the attention of the community, the number of industrial properties has grown from 10 at the end of 1990 to 83 at the end of 2021 (Figure 2), with industrial properties The number represents 7.19% of the World Heritage List, and in the process, the research literature on industrial heritage has gradually increased.

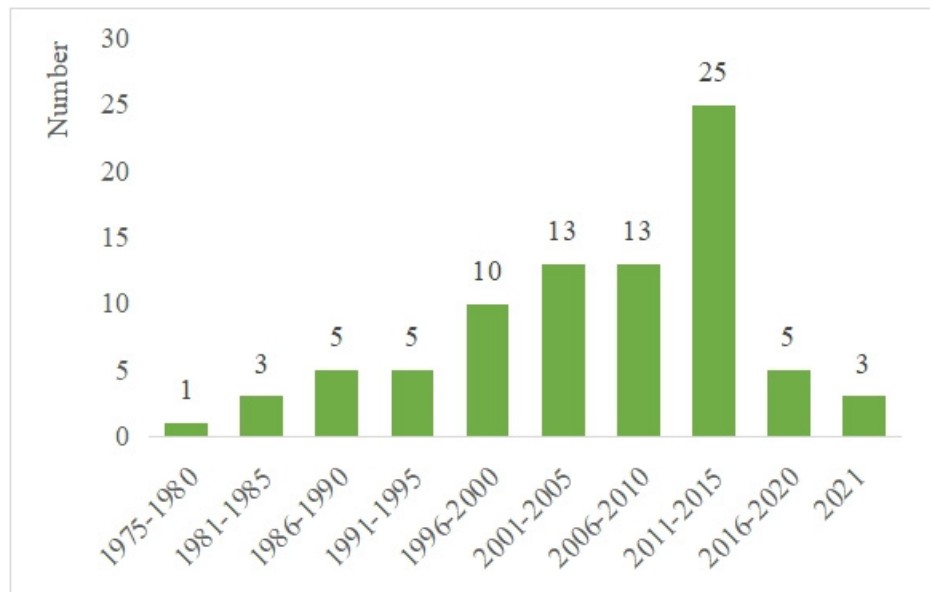

**Figure 2.** Number of industrial properties recognized on the World Heritage List in each year (drawn by authors).

Theoretical research on the regeneration and utilization of industrial heritage began in Western industrial countries. An exact search of the web of science for papers between 1992 and 2021 using the phrase 'industrial heritage' yielded a total of 2323 results. CiteSpace software was used to filter and visualise the search results by region (Figure 3), confirming

that industrial heritage has been studied relatively more in countries with earlier industrial development in the West. The concept of 'industrial archaeology' was formally introduced by the British scholar Michael Rix in 1955. In the 1970s, the First International Congress on the Conservation of Industrial Monuments (FICCIM) was held, and the International Committee on the Conservation of the Industrial Heritage (TICCIH) was established. Since then, the investigation and conservation of industrial heritage in the world gradually increased [5]. The 12th General Assembly of the TICCIH was held in Russia from July 10 to 17, 2003, and adopted the Nizhny Tagil Charter for the Industrial Heritage, which is the most important international charter for the preservation of industrial heritage [6]. In terms of practice, it initially took a conservation-oriented approach with light interventions, such as creating museums and urban public spaces. The conversion of the old malt factory in Snape into the Maltings Concert Hall in 1967 in Suffolk, England, and the conversion of the Seattle Gas Works into a park in 1972 in the United States are examples of static preservation and the reuse of industrial heritage. After years of practical exploration, Western countries improved the regulations and management systems for the reuse of industrial heritage (Figure 3), and gradually paid attention to the enhancement of vitality and the exploration of multi-faceted values in the process of reusing industrial heritage. The 'adaptive reuse' approach to industrial heritage has been gradually promoted by governments and scholars in various countries.

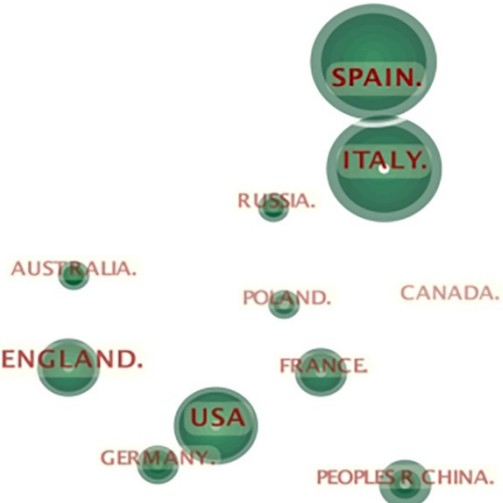

**Figure 3.** Search the Web of Science database for papers from 1992 to 2021 using the subject term industrial heritage, and visualize the regional distribution of the search results using CiteSpace (drawn by authors).

Because of China's relatively late industrial development and long period of industrial development history, the research on the reuse of industrial heritage in China also developed later than that in western countries and did not form a system until the twenty-first century, showing a trend that academic research slightly lags behind practice. In the 1990s, China gradually began to pay attention to the conservation and regeneration of industrial heritage, changing the previously weak awareness of industrial heritage conservation and the problem of extensive demolition and construction of industrial heritage. In 2006, the State Administration of Cultural Heritage of China held the Wuxi Forum, adopted the Wuxi Recommendations, a milestone in the history of industrial heritage conservation in China [7]. Shortly afterward, the State Administration of Cultural Heritage of China issued the 'Notice on Strengthening the Protection of Industrial Heritage', which launched the conservation of China's industrial heritage at the national level. Since then, studies on the adaptive reuse of industrial heritage have increased. Papers on the reuse of industrial heritage included in the CNKI database appeared at the end of the 20th century. Lu Shaoming of Tongji University focused on the reuse value of industrial heritage in

urban docklands, with an emphasis on the study of industrial heritage in urban waterfront areas [8]; Wang et al. put forward scientific proposals for the conservation and adaptive reuse of industrial historical buildings, many of which are located in Chinese cities, in the process of revitalizing old industrial bases and reusing urban architectural heritage [9].

However, even though China has now adopted a policy of graded conservation, the reuse of industrial heritage in China still shows a polarized trend. For industrial heritage listed as national key cultural heritage protection units, China generally adopts a completely untouched conservation approach or conducts appropriate tourism development, with low reuse rates. In contrast, there are parts of industrial heritage that are overexploited in the process of reuse, to the point of being damaged in different senses.

As of June 2022, China's Ministry of Industry and Information Technology published a total of 194 national industrial heritage sites in four batches (Ministry of Industry and Information Technology of the People's Republic China: http://www.gov.cn/zhengce/zhengceku/2021-12/14/content_5660692.htm (accessed on 10 May 2022)). The most famous of these are the Yellowstone Park heritage of the old abandoned Hanye Ping coal mine plant and mine, the Daye iron ore sinkhole and the old Huaxin cement site, as well as the industrial heritage of the Third Line Construction, i.e., 1945 large and medium-sized defence and military industrial enterprises and research institutions built between 1964 and 1983, and nearly 300 small Third Line enterprises built along the border and coast, all of which are being attempted for World Heritage. These industrial heritages are all being renewed for World Heritage inclusion, but have not yet been successful. As can be seen, there is an extremely large amount of industrial heritage in China, but the geographical unevenness of economic development and design levels led to the better renewal of industrial heritage in the more economically developed regions. For example, Shanghai, as the centre of gravity in China's modern national industry, has factory buildings mostly located along the banks of the Huangpu River and Suzhou River, and there are many cases of successful renovation, such as the Minsheng Wharf established in 1956, which in the old days carried grain in and out. After a long period of abandonment and closure, it was finally reopened and renewed in 2016 as a public space for ecological leisure, landscape recreation and public art displays, and its 80,000-ton silo was transformed into an exhibition hall.

Then, this study used CiteSpace software to continue to filter and analyse 2323 documents on the topic of 'industrial heritage' from 1990 to 2021 (removing duplicates and irrelevant (accessed on 10 May 2022)), and obtained five keywords related to industrial heritage (Figure 4). As can be seen from the Figure 4, the keyword 'sustainable development' emerged from 2019, which means that the number of studies on industrial heritage in the direction of sustainable development continued to rise from 2019, and the number of studies on industrial heritage in the direction of sustainability began to increase significantly and rapidly. This indicates that research on the sustainable development of industrial heritage is in line with the needs of the times and is necessary.

**Top 5 Keywords with the Strongest Citation Bursts**

| Keywords | Year | Strength | Begin | End | 1992 - 2021 |
|---|---|---|---|---|---|
| politics | 1992 | 5.64 | **2013** | 2017 | |
| landscape | 1992 | 3.74 | **2014** | 2016 | |
| water | 1992 | 4.54 | **2016** | 2018 | |
| heritage tourism | 1992 | 4.33 | **2017** | 2019 | |
| sustainable development | 1992 | 4.33 | **2019** | 2021 | |

**Figure 4.** Burst keywords of relevant literature with 'industrial heritage' as the subject word in Web of Science analyzed by CiteSpace, 1922–2021 (drawn by authors).

## 2.2. A Review of Low Maintenance Sustainable Design Research

After the industrial age gave way to the information age, the development and application of computer technology affected human lifestyle and thinking. With the continuous progress of science and technology, industry was gradually replaced by emerging electronic

technology industries, such as digital technology, which combines Internet of Things technology, communication technology, artificial intelligence technology, large data technology, and so on. The resulting digital economy became the mainstay of development. In the information age, people no longer pursue basic construction at the cost of the environment, but gradually pay attention to the living environment, environmental quality and environmental friendliness.

The theory of sustainable development originates from ecology and is based on people's concerns about the increasing destruction of the natural environment. The concept of 'sustainable design' has no definite definition in academic fields, but it is closely related to the concepts of 'green design', 'ecological design', 'low carbon design' and 'environmental design' on the one hand, and has its own characteristics and core methods on the other. The Brundtland Report coined one of the most commonly cited definitions of sustainable development in 1987 as "development that meets the needs of the present without compromising the ability of future generations to meet their own needs" [10]. The seminal work that introduced environmental considerations into the world of designers is credited to Victor Papanek's book Designing for the Real World: Human Ecology and Social Change [11].

Combining the findings of Fabrizio Ceschin and Dong Li, the development of the concept of 'sustainable design' can be summarized in four stages [12,13]. The first stage began in the 1980s to 1990s and can be considered as the early Green Design stage. At this time, emphasis was placed on the use of materials and energy with a low environmental impact, focusing mainly on the 3R concept, i.e., Reduce, Recycle, Reuse, with the aim of reducing material and energy consumption and encouraging the easy sorting of products for recycling and reuse. The second stage is the Eco-design stage, which is about the Product Life Cycle design approach to reduce the environmental impact of products throughout their life cycle [14], which can be called 'in-process intervention'. The third stage can be called the eco-efficiency-based 'product-service system design' stage, which goes beyond the general focus on 'physical products' and enters the field of 'system design' [15]. The fourth stage is concerned with social equity and harmony, involving the sustainable development of local culture and the promotion of sustainable consumption patterns [13]. In general, sustainability is a strategic design activity to build a 'sustainable solution' to meet specific needs, reduce the waste of resources and environmental pollution, and to change the quality of people's social life as the ultimate goal.

Since the implementation of sustainability requires indicators in specific contexts of time and space, it is impossible to achieve all the desired outcomes [16]. Sustainable design from a low maintenance, low cost perspective is one of the key entry points and an effective strategy for achieving sustainable design. This study conducted a more in-depth search of subject terms from the sustainable perspective, searching for relevant subject terms in CNKI's Chinese database and the global database Web of Science, resulting in the following table. It was found that there is a scarcity of research in the low-maintenance direction and in the direction of the secondary renewal of industrial heritage in the sustainable field (Table 1).

**Table 1.** CNKI and Web of Science related subject word retrieval data (drawn by authors).

| Subject Word | Web of Science | CNKI |
|---|---|---|
| (Industrial Heritage) | 2323 | 3835 |
| (Sustainable Design) | 54702 | 1201 |
| (Industrial Heritage) AND (Sustainable) | 276 | 126 |
| (Industrial Heritage) AND (Renewal) | 52 | 558 |
| (Industrial Heritage) AND (Reuse) | 130 | 875 |
| (Low-Maintenance Design) | 612 | 66 |
| (Low-Maintenance Design) AND (Sustainable) | 52 | 15 |
| (Industrial Heritage) AND (Secondary Renewal) | 0 | 2 |

Low maintenance implies that from beginning of the design, a low maintenance guarantee is provided for future processes such as construction, usage, and renewal. Post-care of low maintenance projects is an important way to maintain the benefits of the project. There are fewer theoretical studies related to low maintenance in Western countries, and the concept of low maintenance sustainability is mainly demonstrated through practical cases [7]. Chinese scholars' research on low maintenance mainly focuses on material, energy and human inputs as an entry point to study the benefits of the compensation mechanism and reductions in resource consumption [12]. The strategies proposed by Chinese scholars for low maintenance design mainly focus on scientific plant selection, recycling rainwater, reasonable use of energy, economic value creation and the reasonable planning of later labor inputs [12,17]. In the field of information technology, in order to reduce human maintenance and management costs and to achieve a green and low-carbon economy, smart cities are often established with the help of information technology, attempting to establish green and intelligent physical environment detection systems [18] in various areas, which can more accurately capture and focus on issues such as urban sustainability caused by climate or technological change. At the same time, green and smart infrastructure [19] are developed to create sustainable smart eco-cities with a smart and sustainable eco-design [20].

*2.3. Research Gapa*

By conducting a review of past research and practices, we found that research on sustainable development in China and Western countries is widely available, and research on low maintenance design practices in the process of sustainable development is rare. In the CNKI database, 65 papers were retrieved with the subject term 'low maintenance design', and a keywords co-occurrence analysis was conducted (Figure 5), which revealed that most of the relevant research focuses on low maintenance research in landscape design. The researchers are mainly scholars and practitioners in the landscape architecture industry, and the research objects are mostly landscape construction projects. They aim to promote the sustainable and healthy development of landscape architecture with their research results. The practice and theory of the reuse and renewal of industrial heritage itself has matured, but there is a gap in research on how to apply low maintenance sustainable design to the renewal of industrial heritage.

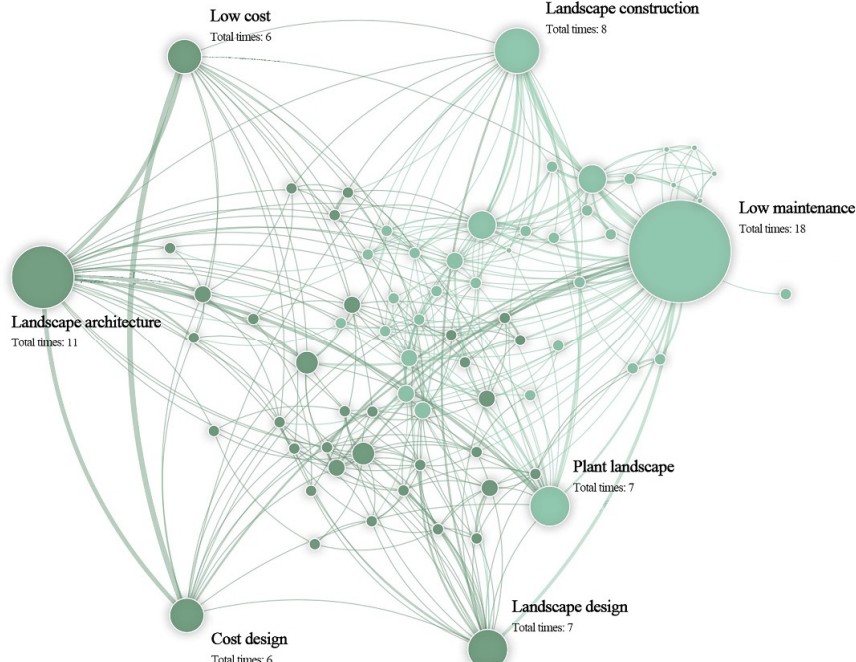

**Figure 5.** Keywords co-occurrence analysis in CNKI database with low maintenance design as the subject term (drawn by authors).

A visual analysis of the frequency of keywords in the literature revealed the top 20 keywords, which are shown in Figure 6. This indicates that the number of studies related to industrial heritage is much greater for conservation and tourism development, and lower for sustainable design and renewal.

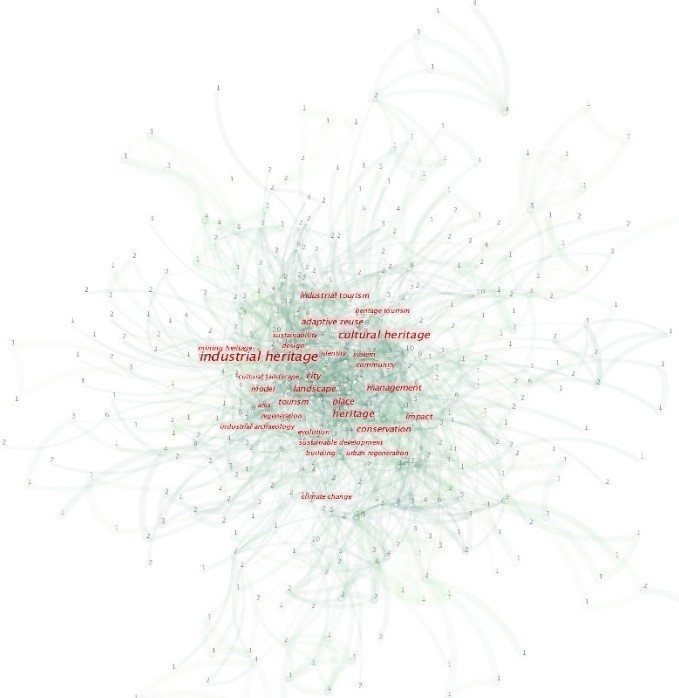

| Keywords | Frequency |
|---|---|
| industrial heritage | 254 |
| culture heritage | 115 |
| heritage | 56 |
| city | 47 |
| management | 38 |
| conservation | 37 |
| adaptive reuse | 36 |
| landscape | 34 |
| place | 34 |
| tourism | 29 |
| industrial tourism | 28 |
| impact | 27 |
| model | 23 |
| sustainability | 20 |
| community | 19 |
| system | 19 |
| area | 18 |
| sustainable development | 18 |
| heritage tourism | 17 |
| identity | 16 |
| evolution | 15 |
| design | 14 |
| urban regeneration | 14 |
| climate change | 13 |
| industrial archaeology | 13 |
| regeneration | 13 |
| building | 13 |
| cultural landscape | 13 |
| mining heritage | 13 |

**Figure 6.** High frequency keywords analysis in the field of industrial heritage research (search the Web of Science database for papers from 1992 to 2021) (drawn by authors).

### 2.3.1. The Lack of Research on Secondary Renewal of Industrial Heritage

After a long process of industrial heritage reuse, the results of earlier industrial heritage reuse were collected for more than ten years or even decades. A literature search revealed that there is a relative absence of research on the aftermath of industrial heritage reuse. Similarly, there is a lack of subsequent sustained attention by scholars on the instances of industrial heritage that have undergone reuse. This situation resulted in limitations and shortcomings arising from the renewal of industrial heritage that was carried out earlier.

After the idea of sustainable development was proposed and gradually elevated to the policy level, a huge amount of literature related to sustainable design was published. Even so, the sustainable development of industrial heritage is still in the intermediate stage. There is a lack of systematic and critical research on the aforementioned phenomenon of industrial heritage. Most studies still concern industrial heritage development, industrial heritage value evaluation, and industrial heritage reuse strategies, and there are few follow up studies on industrial heritage reuse.

In recent years, scholars gradually emerged in China to evaluate the post-use situation of industrial heritage and to construct evaluation systems [21]. Similarly, the sustainability evaluation of industrial heritage reuse [22] and comprehensive evaluation of satisfaction after industrial heritage reuse [23] were conducted, but had little impact on the sustainability development of industrial heritage reuse.

### 2.3.2. The Lack of Research on the Application of Sustainable Design in the Renewal of Industrial Heritage

Globally, research and practices for the first renewal of industrial heritage has matured, but the results for the reuse of industrial heritage after years of use may have limitations.

However, at the same time, there are only a few studies available on the secondary and superimposed renewal of industrial heritage. Only a few of these focus on the establishment of evaluation systems for the use of the built environment after the renewal of industrial heritage, explore the actual participation of people, and reflect on the extent of government participation and the way in which it is carried out [24,25], but the literature generally lacks reflections on the design methods of primary renewal. At the same time, there is a lack of systematic operation strategies for secondary renewal and future superimposed renewal.

This research focuses on the Jinan Tianhong Community Industrial Heritage Park, which previously underwent a renewal, and is now facing the problem of renewal once again ten years after it first renewal. The main focus of this study was to consider future regeneration while conducting secondary regeneration to achieve a sustainable, low cost and eco-friendly regeneration process.

### 3. Method: Secondary Renewal Method under the Guidance of Low Maintenance Sustainability

After the first renewal, the existing site still has many problems, including: the sense of the existence of historical buildings being low, and most of the renewal methods in the past are mainly hidden (Figures 7 and 8). As a result, the existing industrial heritage tanks cannot meet the needs of the times and lack usage, are gradually excluded by modern buildings which leads to the extinction of the original site's spirit of the industrial age. Additionally, the existing sites only have playground facilities, facilities have aged, lack waiting areas, sales, trash cans, shading, pet care and other service facilities, and cannot meet the needs of all ages. The terrain inside the base is relatively flat and there is no flood discharge or storage pool. Rainy days result in inadequate rainwater collection and timely drainage.

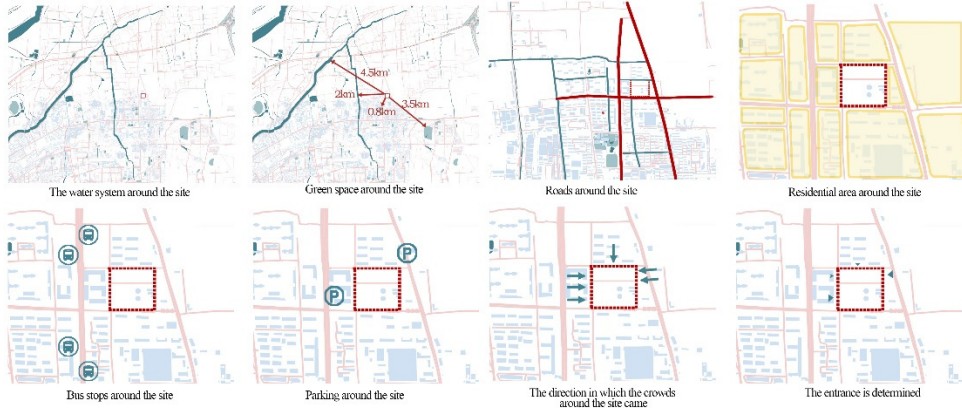

**Figure 7.** Analysis of the site's surroundings (drawn by authors).

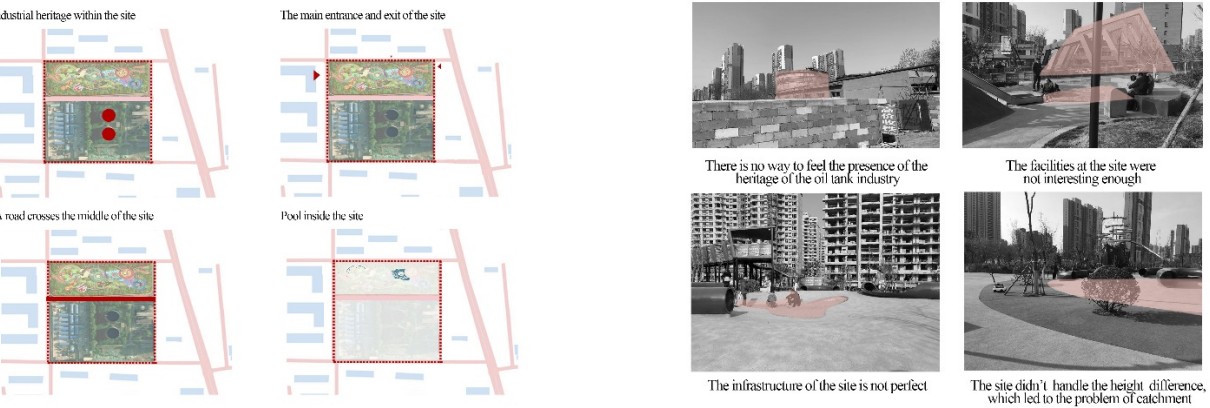

**Figure 8.** Analysis of the internal environment of the site (drawn by authors).

This study used a mixed-method approach to analyse the existing heritage elements within the site, and proposed eight strategies for the second renewal of industrial heritage parks based on the design concept of 'respecting nature and putting ecology first' and the principles of low intervention, low consumption, low maintenance and sustainable development.

This study is based on the 'ecological service' (ES) index system in the composite index system of sustainable development when proposing strategies. After the United Nations Conference on the environment and development in 1992, a theoretical and methodological system of sustainable development with four university research directions was formed in the frontier research field of international theory, namely, ecology, economics, sociology and systems. In 2001, Yang Duogui, Chen Shaofeng and Niu Wenyuan of the Chinese Academy of Sciences summarized the index systems derived from these four directions [26], namely, the ecological 'ecological service' index system (ES), the economic 'national wealth' index system (MW), the sociological 'human development' index system (HD) and the systematic 'sustainability' (SC) index system.

This study selected the ecological direction of sustainable development as the theoretical basis. As a result, we considered ecological balance, natural protection, environmental pollution prevention and control, rational development and the sustainable utilization of resources as its most basic research content. Its focus was 'achieving a reasonable balance between environmental protection and economic development' as an important indicator and basic means to measure sustainable development. The most representative indicator system in this direction is the 'ecological service' (ES) indicator system proposed by Constanza et al in 1997 in the journal *Nature*. For the first time, they systematically designed the 'ecological service indicator system' (ES) to measure the value of the services provided by the global natural environment to human beings. He divided the "ecological service" functions provided by the global ecosystem into 17 types [27]. According to the actual conditions of the cite, this study selected eight elements with development and transformation potential from these 17 ecological service functions, namely water, materials, terrain, plants, heat, wind, bionic structures and industrial building transformation, and then proposed transformation and development strategies (Table 2).

**Table 2.** Selection of update strategy based on ecosystem service (drawn by authors).

| Ecosystem Service | Examples | Strategic in Site | Description |
|---|---|---|---|
| Soil formation | Weathering of rock and the accumulation of organic material. | Terrain treatment | The balance of earthwork and the natural slope angle. |
| Pollination | Provisioning of pollinators for the reproduction of plant populations | Plants | Low maintenance plant selection. |
| Biological control | Keystone predator control of prey species, reduction of herbivory by top predators. | | |
| Refugia | Nurseries, habitat for migratory species, regional habitats for locally harvested species, or over wintering grounds. | | |
| Food production | Production of fish, game, crops, nuts, fruits by hunting, gathering, subsistence farming, or fishing. | | |
| Genetic resources | Medicine, products for materials science, genes for resistance to plant pathogens and crop pests, ornamental species (pets and horticultural varieties of plants). | | |
| Raw materials | The production of lumber, fuel, or fodder. | Materials | Use of energy efficient materials. |

**Table 2.** *Cont.*

| Ecosystem Service | Examples | Strategic in Site | Description |
|---|---|---|---|
| Gas regulation | Co2/o2 balance, o3 for uvb protection, and sox levels. | Heat, wind | The use of solar energy Use natural air for ventilation circulation. |
| Climate regulation | Green-house gas regulation, dms production affecting cloud formation. | | |
| Disturbance regulation | Storm protection, flood control, drought recovery, and other aspects of habitat response to environmental variability mainly controlled by vegetation structure. | Water | Build a water circulation system. |
| Water regulation | Provisioning of water for agricultural (e.g., irrigation) or industrial (e.g., milling) processes or transportation. | | |
| Water supply | Provisioning of water by watersheds, reservoirs, and aquifers. | Bionic structures | Design of a play tower based on the combination of bionics with thermal energy use and water circulation systems. |
| Erosion control and sediment retention | Prevention of loss of soil by wind, runoff, or other removal processes, storage of silt in lakes and wetlands. | | |
| Nutrient cycling | Nitrogen fixation, n, p, and other elemental or nutrient cycles. | | |
| Waste treatment | Waste treatment, pollution control, detoxification. | | |
| Recreation | Eco-tourism, sport fishing, and other outdoor recreational activities. | Industrial building renewal | Integration of industrial heritage and nature. |
| Cultural | Aesthetic, artistic, educational, spiritual, and/or scientific values of ecosystems. | | |

The low maintenance idea of this study starts from the eight aspects and gives corresponding measures to achieve low maintenance by using new technology, new materials and generating relationship with the city. The low maintenance measures in the site can create an impressive scene and atmosphere for visitors, ensuring that the spirit of the original industrial heritage place and the ecological concept of sustainability coexist.

### 3.1. Strategy 1: Build a Water Circulation System

The first step of this strategy was linking the park water system with the urban water system and the surrounding park water system to form an 'urban-park' water circulation system, ensuring the vitality and cleanliness of the water within the park, and linking the park to the urban parkland system (Figures 9 and 10).

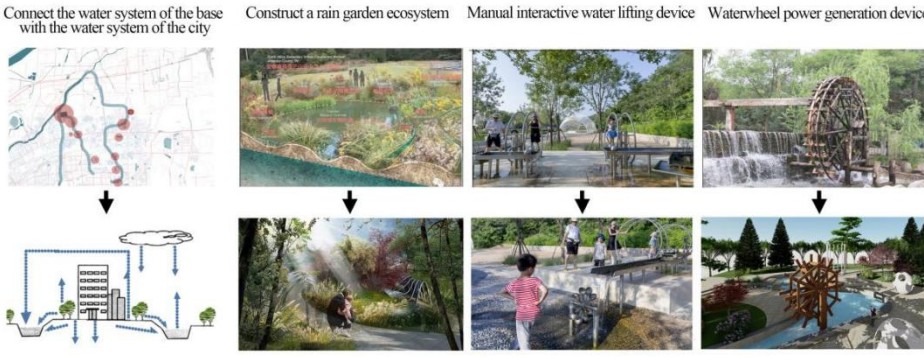

**Figure 9.** Construction method of water circulation system (drawn by authors).

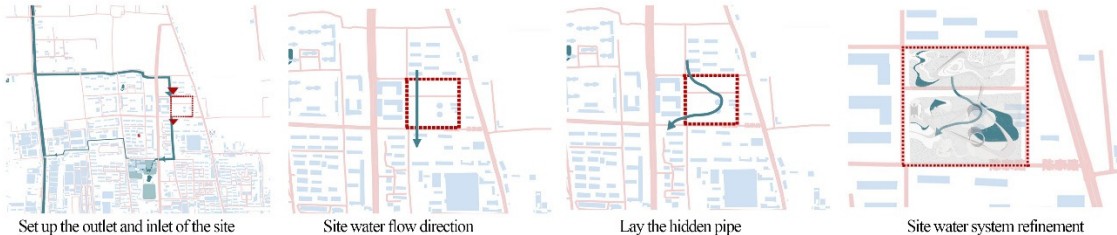

| | | | |
|---|---|---|---|
| Set up the outlet and inlet of the site | Site water flow direction | Lay the hidden pipe | Site water system refinement |

**Figure 10.** Connection between the site and urban water circulation system (drawn by authors).

Secondly, the topography and the designed slope of the ground were used to form a rain garden [28], underground pipes were laid to direct the flow of water from the park to the rain garden, and rainwater storage and secondary recycling devices were established to ensure the infiltration, storage, filtration and recycling of rainwater on site through the rain garden, while ensuring the effective recycling of sewage [29] for greening and watering [30], car washing and road washing (Figure 11).

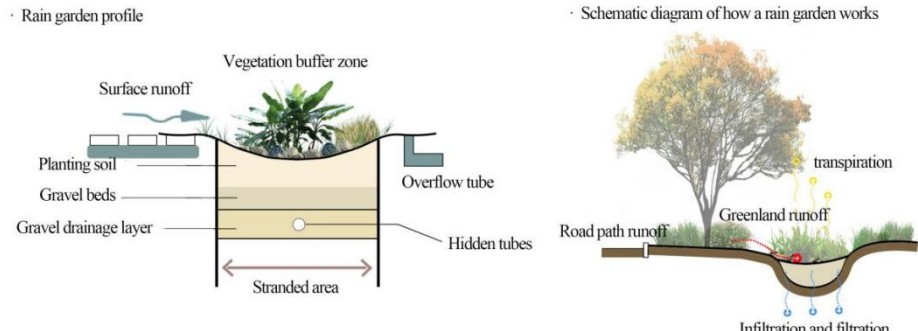

**Figure 11.** Rain garden (drawn by authors).

Thirdly, the interactive water lifting device was combined with the irrigation system, using the kinetic energy generated when people use the facility to drive the water circulation system, either to form a fountain or waterfall landscape, or for irrigation.

Fourth, the height difference between the site's water diversions and the city's waterways was used to form a stack of water, driving the water circulation and driving the waterwheel, which in turn generates electricity and provides power.

Fifth, roof siphon drainage systems on landscape structures can be used, by using the difference in air potential energy at different heights to make a large local vacuum inside the siphon pipe drainage system, thus achieving the rapid discharge of siphoned rainwater through the action of siphoning.

Sixth, the use of a ground source heat pump system [31]. Taking advantage of the geographical characteristics of Jinan, that is, some physical and chemical characteristics of relatively stable rock temperature, such as rock, soil and so on, can effectively reduce the energy consumption of underground minerals and other energy sources, so as to make the system carry out rapid electric energy and heat exchange with other energy systems, such as rock, soil and rock, soil and so on. In this way, the heat source and energy can be transmitted quickly. The system can be combined with the outdoor swimming pool. The buried pipe ground source heat pump system combined with water-cooling multi connected mode was adopted to cool and heat the outdoor swimming pool on the site by extracting the cold and heat of the soil, and deliver domestic hot water at the same time.

### 3.2. Strategy 2: Choose Energy Efficient Materials

To achieve low maintenance and sustainability, alternative, durable, easy to clean and anti-pollution, robust materials should be chosen for space creation.

Eco-permeable bricks or grass-planting bricks should be selected for surface paving, and eco-road materials should be used at the same time to improve the performance of

breathing, absorbing heat and penetrating rainwater on the pavement. For example, when pervious concrete was used, it consisted of cement paste, coarse aggregate and voids, without fine aggregate, and only a thin layer of cement paste on the surface of coarse aggregate. The voids were large, and the permeability was controlled by the voids, while the strength depended on the coarse aggregate and the cement. This material is mostly used in roads, parking lots, pedestrian streets and other places, which can effectively solve the following problems:

- Municipal drainage, urban waterlogging and excessive water accumulation on the road surface, and ensure the circulation of surface water and the storage of underground water sources;
- Reduce noise pollution;
- Ensure the permeability, water retention and air permeability of the ground;
- Adjust temperature, improve air quality and regulate ecological balance.

The material of exterior elevation that was selected was corrosion-resistant and aging-resistant plastic steel instead of non-ferrous metal. Translucent acrylic was used to create light and shadow capturers that create space, while using materials that constantly create different aesthetics over time, such as increased decoration with corroded steel plates that can vary over time. In addition, an ecological board can be selected. Its properties are composite straw board. Its basic raw materials are natural mineral stone powder and natural fibers. Under the application of inorganic gelling technology, new materials were obtained by molding and natural curing, and its natural and environmental protection features ensured its safety in use (Figure 12).

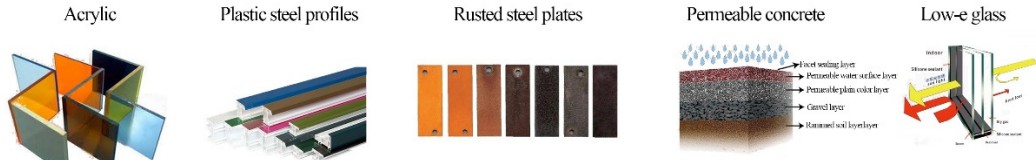

**Figure 12.** Energy-saving materials (drawn by authors).

A diatom mud ecological wall was selected as the wall surface of landscape structures [32]. Latex paint contains many harmful gases and the adhesive of wallpaper has many harmful components. Diatom mud has strong light absorption and does not cause light pollution. Furthermore, it prevents static electricity and dust hanging. It has a good sound-absorption and thermal-insulation effect and can be fire-retardant.

Low E glass, a coated glass with high reflectivity to infrared rays ranging from 4.5 μm to 25 μm wavelengths, was selected for buildings and landscape structures with low radiation [33]. It reflects infrared heat energy indoors in winter, guarantees indoor temperature, absorbs a lot of light from cement roads, buildings' exterior walls, and sunlight in summer, and radiates around in far infrared. It has the effect of anti-glare, one-sided perspective and improved comfort. In addition, heat-absorbing glass was selected, which uses metal ions in the glass to selectively absorb solar energy, showing different colors and generally reducing solar radiation heat by 20–30%. The color and transmittance ratio of glass can be changed according to the composition and concentration of metal ions in the glass, and can be combined with the Eco-tower to create different light atmosphere.

### 3.3. Strategy 3: Low Maintenance Plant Selection

In the selection and treatment of plants, priority should be given to plants that are native, participatory and have a low management costs [34], and then low maintenance should be achieved through natural landscaping and community-based planting, while child-friendly plants and plants commonly used in rain gardens should be selected to create an ecological place of co-prosperity and symbiosis.

- Choose plants that are native and have low management costs. Native plants are easy to source, inexpensive to purchase, have a high survival rate, require less maintenance, less irrigation, pruning and management costs at a later stage, while choosing plants with a long growth cycle, high survival rate and pruning resistance. Avoid using a large number of grasses and potted flowers [35].

- Choose participatory plants. Social participation in greenery planting activities can indirectly reduce maintenance costs for managers. Public welfare landscape forestry which is easy to maintain can better attract social participation, such as economic fruit trees, not only for people to enjoy, and their fruits also have a certain economic benefit, so the costs saved can be used for later plant maintenance, and it indirectly reduces the cost of landscape maintenance.

- Ensure community planting, by trying to create a small lawn, a moderate number of trees and a sufficient number of shrubs. The plants in the community are more viable and require lower late-management costs. In the construction of the plant community, we should insist on the combination of trees, shrubs and grasses, with trees being the main design plan. Trees have a longer growth cycle and are easier to maintain. In order to create a spatial hierarchy and walking experience in dense forest areas, the plant configuration should be richly layered, enclosing the space with the help of plants and distributing them with flowering and fragrant plants to stimulate people's desire to explore, such as using soapberry, loblolly pine, elm and cedar for trees, pearl plum, red raspberry, purple magnolia and golden silver wood for shrubs, and paving loblolly pine, iris, maidenhair and cassia for ground cover plants.

- Choose plants with a strong self-cleaning ability, such as an outer barrier plant configuration to absorb harmful gases, dust and noise-blocking plants, for trees choose big-leaved maidenhair, acacia, cedar and long-clawed acacia, for shrubs choose golden-leaved maidenhair, sea tree, zinnia and purple yucca, for ground-cover plants choose pavement loblolly pine, iris, maidenhair and cassia. Secondly, the use of wetland landscapes, which are relatively stable due to the complexity of the system, requires little human intervention as they are self-maintaining and can be used more often when available. The plant configuration of the ecological wetland area is based on plants that are resistant to water and humidity, such as weeping willow, sequoia, red maple and peach trees for trees, and aquatic iris, Chiffchaff, lotus, water lilies, reeds, cattails and water bamboo in terms of aquatic plants.

- The use of green roofing techniques [36] on landscape structures also enhances roof insulation and soundproofing. Choose light-loving, temperature-resistant, cold-resistant, heat-resistant, drought-resistant, barren and vigorous flowering plants to achieve a staggered height of plants and flowers. The main plant species include red-leaved heather balls, small-flowered gardenias, small-leaved boxwoods, golden-sided boxwoods, Hellebore balls and self-purchased finished potted plants, and select vine plants to increase ornamental properties.

### 3.4. Strategy 4: Rational Terrain Treatment for Energy Recycling and Low Maintenance Costs

Firstly, the balance of earthwork [37], excavation and filling in the garden is an important step in reducing the amount of work and maintenance later on.

Secondly, the natural slope angle is used to allow rainwater to flow into the ground, effectively replenishing groundwater and alleviating some of the urban environmental problems such as the sharp drop in the water table in the city, effectively eliminating the environmental pollution hazards such as oil compounds on the ground, while protecting groundwater, maintaining the ecological balance and alleviating the urban heat island effect [38].

Thirdly, the construction of a depressed green space below the road elevation is an option. The planting of water-resistant plants in the depressed green space allows surface runoff water to converge into the space, forming patches of saturated water, delaying the retention time of shallow water and facilitating the water fixation and retention role of

the plant root system, which is conducive to the growth of plants and the purification of groundwater, as well as reducing artificial irrigation. It can beautify the landscape, regulate runoff and purify rainwater through filtration and adsorption.

Fourthly, plant roots are used to reinforce the waterfront, prevent erosion and achieve sustainability in waterfront maintenance.

Fifthly, space should be reserved for expansion and renewal for future regeneration plans and phased planning should be carried out to reduce wasteful renewal, demolition and redevelopment costs in the long term (Figure 13).

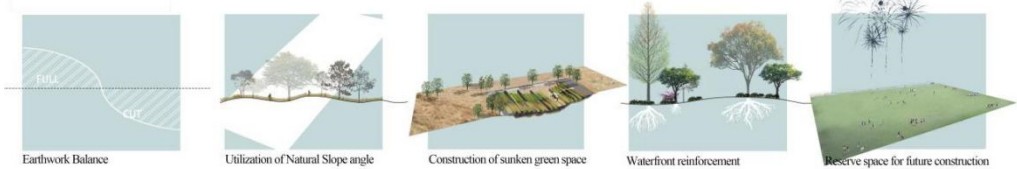

**Figure 13.** Strategies for terrain use (drawn by authors).

### 3.5. Strategy 5: The Use of Solar Energy

The design and shape of the landscape structures and resting seats in the park were combined with solar panels and lighting facilities, and the collected solar energy can power the facilities in the park through solar thermal storage devices, while using insulation and shading materials, solar panels and solar lamps to create solar-power generation and a heat-storage system for low maintenance.

The external surface of the building can be replaced with solar panels and solar photovoltaic glass to create a solar photovoltaic system [39] which consists of solar collectors, a volumetric heat exchanger, a solar mass water tank and solar collector circulation lines (Figure 14). Photovoltaic power generation on the roof and the installation of photovoltaic power generation systems on play towers can be implemented, as this electricity can be generated instantly. Using the water-repellent nature of the glass material of the photovoltaic module, the photovoltaic module is combined with a roof surface with a certain slope and supporting guttering elements to form a rigid waterproof structure. This maintains the waterproof and thermal insulation function of the roof surface, while at the same time it increases the power generation function of the roof surface and enhances the aesthetics of the landscape structure.

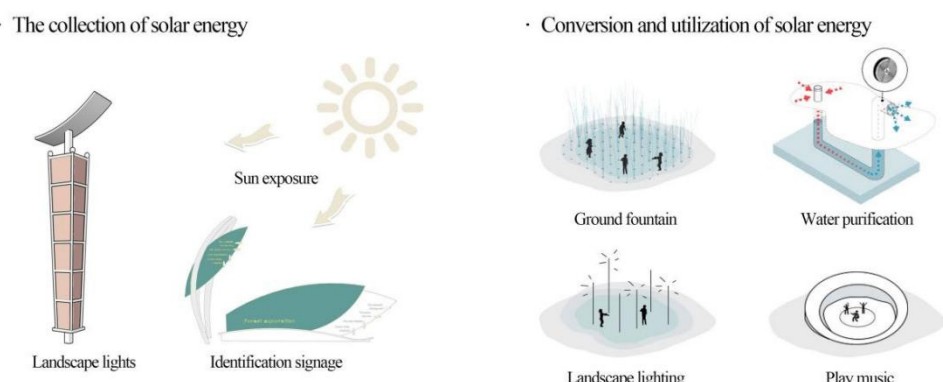

**Figure 14.** How solar energy is collected and utilized (drawn by authors).

### 3.6. Strategy 6: Use Natural Air for Ventilation Circulation

The natural monsoon was first identified, and then the overall layout was designed to provide a corridor to guide the summer monsoon through the location of the structures and combined with the water landscape to ensure the site is ventilated and cooled [40].

In terms of ventilation within the building, the design of the landscape structure was used to create a horizontal ventilation system and a vertical ventilation system to provide a good ventilation effect at walking height and reduce the use of mechanical ventilation.

In some large sites, where natural conditions allow, a combination of wind and solar power generation can be used to reduce energy consumption.

### 3.7. Strategy 7: Design of a Play Tower Based on the Combination of Bionics with Thermal Energy Use and Water Circulation Systems

In order to create a stimulating, eco-friendly complex that serves as a link and supply station for the 'zero maintenance' of the whole park, the vertical greening can be achieved by using its height and shape for solar energy collection and rainwater harvesting, and decorated with vine plants. The anthill, which has the same spatial function as the embedded model in nature, was refined, reversed and translated (Figure 15), and the modelled functional spaces were embedded in an ecological matrix, with each small space implanted into the structure of the play tower in the form of a modelled capsule, linking the different activity and adventure spaces, such as the stained-glass space and the suspension climbing space. At the same time, the anthill's drainage system and traffic system were translated based on bionics, making the tower a core structure as a zero-maintenance design, forming a 'solar energy collection station', a 'rainwater drainage station' and a 'vertical green sky garden' (Figure 16).

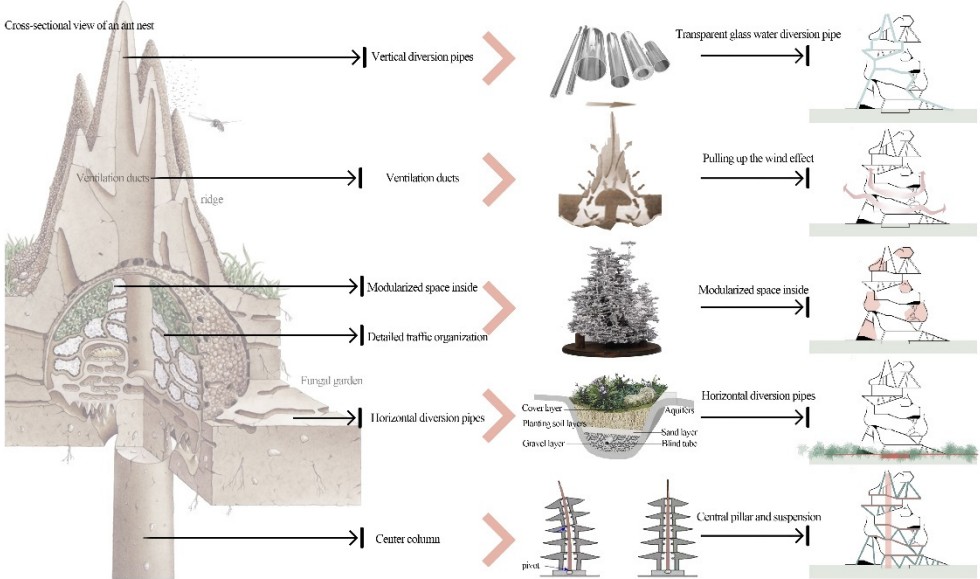

**Figure 15.** Translation from anthill to Eco-tower (drawn by authors).

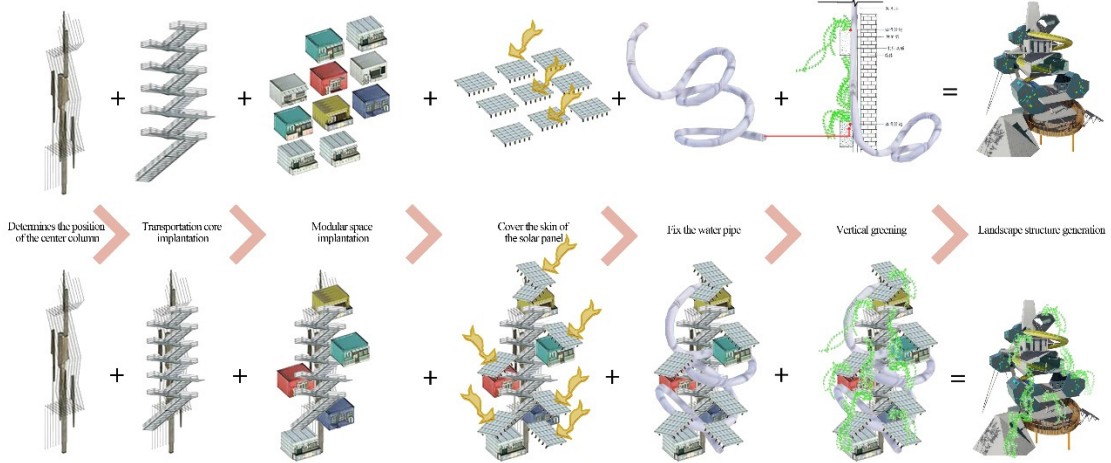

**Figure 16.** The process of constructing an Eco-tower (drawn by authors).

The landscape structure was designed in a polygonal shape and covered with solar panels to collect solar energy for conversion into electricity. Inside, water is collected in a diversion pipe that feeds into the rain garden and underground pipes, and the side walls of the diversion pipe should be planted with vertical greenery. In terms of light use, natural light can be used in conjunction with coloured glass, profiling and special materials to create a spatial atmosphere and guide the play path (Figure 17).

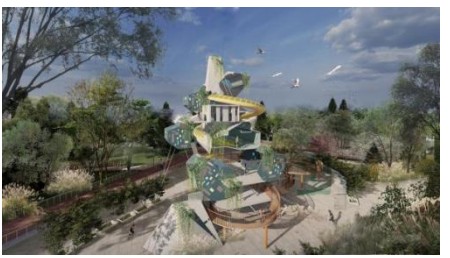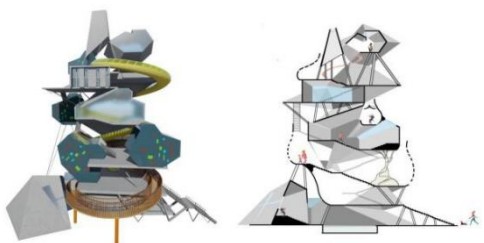

**Figure 17.** Eco-tower rendering, elevation and section (drawn by authors).

### 3.8. Strategy 8: Integration of Industrial Heritage and Nature

The tanks were reconfigured by removing the top and bottom of the tank, leaving only the tank elevation shell, with holes cut into the shell. Water, sand and vegetation were introduced into the internal ground so that the tank was wrapped in its natural environment and well-integrated. When the main road passes through the tank, a 2.5 m slope lift can be carried out to pass through the openings in the tank shell, creating the sensation of walking through the tree canopy, transforming the pedestrian's perspective and better merging the industrial heritage with nature (Figure 18). This approach allows for the industrial heritage to be integrated into the natural landscape, reducing maintenance costs and making it part of the landscape itself, allowing for more possibilities in the subsequent renewal of the superimposed rows.

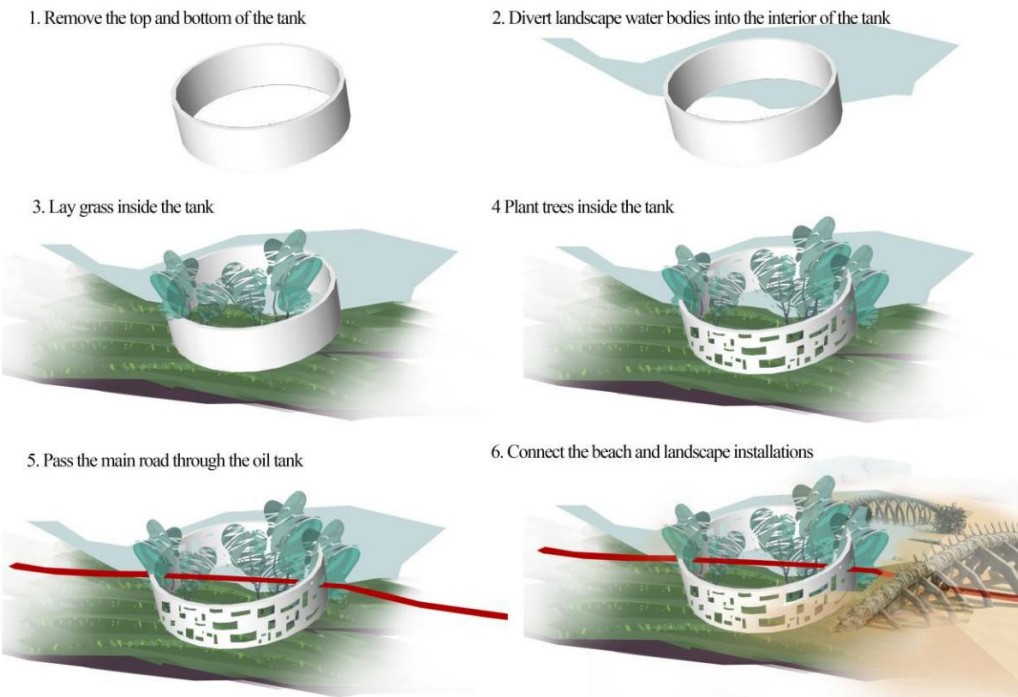

**Figure 18.** Solar photovoltaic glass (drawn by authors).

## 4. Results

The use of the eight renewal strategies mentioned above, in terms of long-term planning, can effectively improve the efficiency of overlapping regeneration, reduce the cost of demolition and reconstruction, provide space for subsequent regeneration, operability, and facilitate the most efficient overlapping row of regeneration in the urbanization process.

At the same time, the strategy takes time changes into consideration, which ensures that sustainability and spontaneous renewal are maintained within the site, such as the self-purification of water, energy reserves, plant growth, and symbiosis between industrial heritage and the landscape, which will make the industrial heritage park continuously produce positive changes in the temporal dimension and reduce the risk of plant humus pollution, equipment ageing, and the abandonment of structures brought about by negative changes, and also provides new ways of storing groundwater sources and promotes the greening of the city (Figure 19).

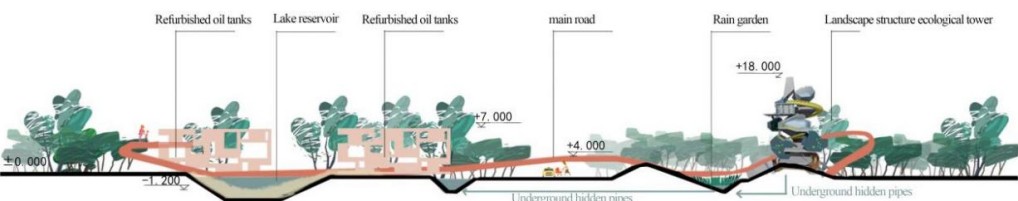

**Figure 19.** Site Profile. (drawn by authors).

The strategy system is based on the disadvantages of the first renewal and the current needs of industrial heritage, and allows for a sustainable renewal process by avoiding material losses and the wasteful demolition and abandonment of industrial heritage, reducing maintenance costs and energy losses in the process of subsequent superimposed renewal.

## 5. Conclusions

In conclusion, under the guidance of the concept of low maintenance and sustainable development, the superimposed renewal and transformation of industrial heritage should be carried out from an ecological perspective.

In order to achieve low maintenance and improve the service life of the project, the operation strategies and techniques of the low cost and low maintenance concept should run through every link of the secondary renewal of industrial heritage, mainly starting from the aspects of water system construction, material selection, plant selection, terrain treatment, landscape structure construction under the concept of energy conservation, ecological treatment of industrial buildings, etc., so as to improve the utilization efficiency of resources and reasonably select landscape plants and facilities. Ultimately, the concept of local adaptation and low maintenance will be combined.

However, China is still a developing country and may face some real economic problems in the early stages of attempting to promote and develop low maintenance sustainable industrial heritage secondary regeneration designs, and may face situations where the budgetary costs for the short term construction of low maintenance designs are greater than normal construction costs. Examples of such problems include some advanced energy efficient materials may be more expensive, more advanced low maintenance scientific pre-design costs are more expensive, and low maintenance sustainable regeneration requires a more sophisticated construction process leading to longer construction periods. These issues raise doubts among investors about the feasibility of low maintenance sustainable design. In order to achieve rapid short-term benefits, they prefer to choose traditional designs with fast construction process and low cost, which prevents the effective promotion and application of low maintenance sustainable renewal design.

At this time, the government can take action in two ways. First, the government, as a public investor, should try to introduce perfect and reasonable compensation policies to make low-maintenance sustainable designs popular. For example, when private investors

undertake projects to update small industrial heritage, they can apply for environmentally friendly compensation, that is, when they use environmentally friendly materials or environmentally friendly designs, they may exceed part of their budget. At this time, they provide the government with corresponding and exact energy-saving analysis data. After feasibility is evaluated by government professionals, the government can provide them with appropriate economic subsidies and rewards, gradually leading to the popularity and development of the design for the secondary renewal of low maintenance sustainable industrial heritage.

Secondly, the government and public investors should not only consider the speed of construction, the aesthetic degree of construction and short-term economic benefits, but also take long-term economic benefits brought by energy conservation into account and set up a longer-term economic planning vision. The low maintenance of eco-friendly facilities can help to reduce subsequent maintenance and management costs, pollution control costs, water maintenance costs, heat costs, and the consumption of electricity.

**Author Contributions:** Conceptualization, Z.L.; methodology, Z.L. and Q.G.; software, Z.L. and Q.G.; validation, Z.L. and Q.G.; formal analysis, Z.L. and Q.G.; investigation, Z.L. and Q.G.; data curation, Z.L. and Q.G.; writing—original draft preparation, Z.L.; writing—review and editing, Z.L., Q.G. and L.Q.; visualization, Z.L. and Q.G.; supervision, L.Q. All authors have read and agreed to the published version of the manuscript.

**Funding:** This research was funded by Center for Balance Architecture, Zhejiang University. And The APC was funded by Center for Balance Architecture.

**Conflicts of Interest:** The authors declare no conflict of interest.

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
