# Peer review of "Superimposed Renewal of Industrial Heritage under the Guidance of Low Maintenance and Sustainability—Renewal of Refinery Site in Jinan Tianhong Community"

_sustainability, doi:10.3390/su14127486_

Round 1

Reviewer 1 Report

The text is not complete from a scientific perspective. In particular, there are a many issues that should be considered when revising the text, namely:

-      the research context is missing (why was this study born? is a redevelopment project underway?)

-        the literature review is scarce and generic (it also considers a very broad concept of sustainability which appears not so interesting to define the specific theoretical framework of the article)

-        in some parts of the text the bibliographic references are missing, although they are necessary to support the authors' statements (e.g. for lines 197-199)

-        the authors never provide a definition of “industrial heritage”, that is the main issue of the paper (does any abandoned industrial area become a heritage to be recovered?)

-        in the conclusions and results, a reflection on the economic feasibility of the project lacks. Which actor should take care of this sustainable renewal? A public or a private investor? With what economic resources? Moreover, there is not any reflection about the limits of the project in terms of sustainability.

Author Response

Dear  Professor,

Thank you very much for your letter and advice. We have revised the paper, and would like to re-submit it for your consideration. And we would like to express our sincere thanks to your constructive and positive comments.

We have addressed the comments and hope that the revision is acceptable, and I look forward to hearing from you soon.

With best wishes,

Yours sincerely,

Zijia Li

Reviewer 2 Report

In general, I enjoyed this paper. This case is quite interesting to go further about the superimposed renewal of industrial heritage. However, I suggest better referencing some concepts, for example, the information age.

Author Response

(The authors gave the same response as above.)

Reviewer 3 Report

Dear Authors, 

Thank you for your work which is great student work, at least the structure and level suggest that. 

It needs major changes starting from adequate quoting - there is numerous quotation " " without reference. If you consider those notions like so-called something, then put in ' '. If you quote something then you must indicate the author, date of publication and the page number (if it is not given otherwise). 

Figures: low quality and not referenced. Are those works made by you or found somewhere else? If somewhere else, then do you have permission to use them?

The sentences are often long and not clear - regardless of the grammatical adequacy. 

In the text, you use "some scholars" (p. 3) or "There are fewer theoretical studies related to..." (p. 3) - these are statements without clear and well-proved pieces of evidence and there are more such and similar statements which cannot stand without proper background (references).

You are talking about "low maintenance sustainable superimposed renewal" from the beginning or superimposed design, but it seems that your fictional definition is not well developed... you have to define what you mean under this phenomenon and not just mention that it would be great to apply it. 

It is nice how you put together the strategy but I cannot see the data upon which you came up with these ideas (again are these your inventions come somewhere - there are no references in the text and no sources of the illustrations). 

Overall the manuscript is a great start but plenty of things are missing which are required for such an ambitious work (see above). More on-site research and data collection, not so general conclusions (etc.).

Maybe you did all these... but in the text, it is not visible.

Best wishes

Author Response

(The authors gave the same response as above.)

Round 2

Reviewer 1 Report

Thank you for having accepted the reviews

Author Response

Dear  Professor,

Thank you very much for your time involved in reviewing the manuscript and your very encouraging comments on the manuscript.

We would like to take this opportunity to thank you for all your time involved and this great opportunity for us to improve the manuscript. We hope you will find this revised version satisfactory.

And we would like to express our sincere thanks again to your constructive and positive comments.

With best wishes,

Yours sincerely,

Zijia Li

Reviewer 3 Report

Dear Authors,

you made remarkable changes and the quality of you paper became higher. The only remark this time to follow the template and somehow manage the quality of your images (in Word use jpg and 300 dpi at least). These are formalities but could add a lot to your paper. If these changes are already requested by the side of the editors, please forget them.

Thank you for your work!

Author Response

(The authors gave the same response as above.)
